# Plant Hormone Pathway Is Involved in Regulating the Embryo Development Mechanism of the *Hydrangea macrophylla* Hybrid

**DOI:** 10.3390/ijms25147812

**Published:** 2024-07-17

**Authors:** Yali Zhu, Xiaoman Zeng, Tingting Zhu, Hui Jiang, Penghu Lei, Huijun Zhang, Haixia Chen

**Affiliations:** College of Horticulture, Hunan Agricultural University, Changsha 410128, China; 986zhuyali@stu.hunau.edu.cn (Y.Z.); 18707378532@163.com (X.Z.); ztt1728160041@163.com (T.Z.); jianghui@hunau.edu.cn (H.J.); 15193390259@stu.hunau.edu.cn (P.L.); hjun_zhang@163.com (H.Z.)

**Keywords:** embryonic morphology, transcriptome, differential gene, molecular mechanism

## Abstract

The research is aimed to elucidate the role of plant hormones in regulating the development of hybrid embryos in *Hydrangea macrophylla*. Fruits from the intraspecific cross of *H. macrophylla* ‘Otaksa’ × ‘Coerulea’ were selected at the globular, heart, and torpedo stages of embryo development. Transcriptome sequencing and differential gene expression analysis were conducted. The results showed that fruit growth followed a single “S-shaped growth curve, with globular, heart, and torpedo embryos appearing at 30, 40, and 50 d post-pollination, respectively, and the embryo maintaining the torpedo shape from 60 to 90 d. A total of 12,933 genes was quantified across the three developmental stages, with 3359, 3803, and 3106 DEGs in the S1_vs_S2, S1_vs_S3, and S2_vs_S3 comparisons, respectively. Among these, 133 genes related to plant hormone biosynthesis and metabolism were differentially expressed, regulating the synthesis and metabolism of eight types of plant hormones, including cytokinin, auxin, gibberellin, abscisic acid, and jasmonic acid. The pathways with the most differentially expressed genes were cytokinin, auxin, and gibberellin, suggesting these hormones may play crucial roles in embryo development. In the cytokinin pathway, *CKX* (Hma1.2p1_0579F.1_g182670.gene, Hma1.2p1_1194F.1_g265700.gene, and NewGene_12164) genes were highly expressed during the globular embryo stage, promoting rapid cell division in the embryo. In the auxin pathway, *YUC* (Hma1.2p1_0271F.1_g109005.gene and Hma1.2p1_0271F.1_g109020.gene) genes were progressively up-regulated during embryo growth; the early response factor *AUX/IAA* (Hma1.2p1_0760F.1_g214260.gene) was down-regulated, while the later transcriptional activator *ARF* (NewGene_21460, NewGene_21461, and Hma1.2p1_0209F.1_g089090.gene) was up-regulated, sustaining auxin synthesis and possibly preventing the embryo from transitioning to maturity. In the gibberellin pathway, *GA3ox* (Hma1.2p1_0129F.1_g060100.gene) expression peaked during the heart embryo stage and then declined, while the negative regulator *GA2ox* (Hma1.2p1_0020F.1_g013915.gene) showed the opposite trend; and the gibberellin signaling repressor *DELLA* (Hma1.2p1_1054F.1_g252590.gene) increased over time, potentially inhibiting embryo development and maintaining the torpedo shape until fruit maturity. These findings preliminarily uncover the factors affecting the development of hybrid *H. macrophylla* embryos, laying a foundation for further research into the regulatory mechanisms of *H. macrophylla* hybrid embryo development.

## 1. Introduction

*Hydrangea macrophylla* is an important species of Hydrangea Linn. in the Hydrangeaceae family. China has an abundant germplasm resource of Hydrangea, with 47 species and 11 varieties. In recent years, a series of new varieties has been produced through interspecific or intraspecific hybridization [1]. However, hybridization followed by embryo developmental arrest or even abortion has severely hindered the breeding process of Hydrangea species [2]. The abnormal development of young embryos is mainly because the young embryos may stop developing or die in proembryonic stage, globular embryo, heart embryo, torpedo embryo, cotyledone embryo and other stages [3]. Embryo abortion is mainly caused by abnormal endosperm or disharmony between embryo and endosperm, resulting in endosperm deformity, disintegration, embryo sac deformity, cavity and other phenomena [4]. Therefore, it is of great significance to study the molecular mechanisms underlying the fruit development of *H. macrophylla* hybrids.

Fruit development and seed formation are complex processes, which are not only affected by genetic factors, but also related to environmental conditions and the regulation of plant growth and development, and many genes and transcription factors are involved in the regulation of the processes [5,6]. Plant hormones and transcription factors work together to construct a complex regulatory network that promotes the normal growth and development of fruits and seeds. Studies have shown that plant hormones that play a key role in seed development mainly include auxin (IAA), cytokinin (CK), abscisic acid (ABA), gibberellic acid (GA), ethylene (ETH), brassinolide (BR), jasmonic acid (JA), and salicylic acid (SA) [7]. 

For instance, IAA produced by the central cell/endosperm of Arabidopsis mutants after fertilization will promote the development of seed coat [8]. The content of IAA in maize was low at the early stage of endosperm development, but increased gradually with endosperm maturation [9]. 

Literature results indicate that ABA has dual effects on seed embryo development. In the early embryo development of Arabidopsis thaliana, the increase of ABA expression promoted embryo growth [10]. The decrease or insufficiency of ABA expression in tobacco seed development can lead to seed sterility and inhibit embryo growth [11].

Findings from the literature propose GA can promote seed germination, endosperm nutrition reserve and fruit development. At the late stage of GA biosynthesis, *GA20ox* and *GA3ox* catalyzed the formation of bioactive gibberellin from the interspecific products, while *GA2ox* inhibited the synthesis of GA [12]. In barley endosperm, the downregulation of *HvGA3ox2* and upregulation of *HvGA2oxs* lead to a decrease in GA expression levels [13]. The increased expression of *PsGA2ox2* in Arabidopsis thaliana resulted in decreased ovule number and seed abortion [14]. Bitter melon GA promoted seed germination by increasing protease degradation of RGL2 [15]. GA positively regulated rice can promote α-amylase expression and decrease endosperm starch reserve [16]. 

Studies suggest CK acts on the primary stage of seed development. At the same time, it promoted cell division to form more ovule, and the high expression of genes related to CK synthesis in the early stage of development increased the number of seeds [17]. CK content in spring wheat seed development decreased with seed maturity, and the content in the first two developmental stages was significantly higher than that in the last three developmental stages [18].

Data from literature implies BR can regulate embryonic development and control seed shape. Arabidopsis thaliana regulates embryo and endosperm development and seed formation by directly or indirectly regulating BR gene expression through BZR1 [19]. The content of BR in wheat tad11-2a-1 mutant decreased, which made the seed volume smaller [20]. 

Historical documents support JA is mainly involved in embryo development and seed production during seed development. The decrease of JA-lle expression in rice seed would lead to the decrease of seed yield [21]. The increased expression of JA-lle during early development of barley can promote the reserve of nutrients during embryonic development [22]. 

Academic findings infer SA regulates seed germination and fruit ripening, and plays a role in plant growth and development by regulating cell division and expansion. The decrease and deletion of SA expression during the development of Arabidopsis sid2 mutant led to an increase in seed number [22]. The significant expression of SA in early and late development of barley seeds promoted the division and expansion of embryo and endosperm cells [13].

During fruit and seed development, transcription factors regulate organ formation, cell division and differentiation, and fruit and seed maturation [23]. The overexpression of the FtMYB15 gene extracted from Tartary buckwheat in Arabidopsis resulted in increased pigment and anthocyanin content in the seed coat [24]. The overexpression of the AP2/EREBP family *LaAP2L* gene in Larch can enhance cell proliferation and increase seed yield [25]. The bHLH transcription factor gene *SlPRE2* in tomato was highly expressed in immature fruit, and the seeds in *SLPRE2*-silenced strains were smaller. In addition, the down-expression of *SlPRE2* led to the down-expression of genes related to gibberellin metabolism in immature fruit [26]. The expression levels of *AdbZIP24*, *AdbZIP36*, *AdbZIP37*, and *AibZIP30* decreased gradually during the growth and development of peanut seeds [27]. RIN of the MADS-box transcription factor in tomato can directly activate the bHLH95 transcription factor to regulate fruit ripening [28]. In a watermelon study, the NAC transcription factor *CLNAC68* positively regulated seed development by inhibiting *ClGH3.6* [29].

Transcriptomics is a scientific method to systematically study gene transcription profiles at the transcriptional level. Transcriptomic analysis during embryo development is a widely used strategy to identify functional genes in embryo development. In recent years, this technology has been used to study the seed development mechanism of many plants, such as Arabidopsis [29], tomato [30], and peony [31]. However, transcriptomic analysis technology has not been applied to the study of the seed development of *H. macrophylla*, and genes related to seed embryo development and fruit formation have not been reported. In this study, the dynamic changes in fruit growth and development 0–100 days after pollination were observed using ‘Otaksa’ × ‘Coerulea’ in specific hybrid fruits as materials. Fruits at the large globular embryo stage (30 d after pollination), heart embryo stage (40 d after pollination), and torpedo embryo stage (50 d after pollination) were selected for transcriptomic analysis to screen the differentially expressed genes related to plant hormone bio-anabolism in order to elucidate the molecular mechanism of hormone biosynthesis in the development of *H. macrophylla* hybrid fruits, and lay a foundation for revealing abnormal embryo development after the pollination of *H. macrophylla*.

## 2. Results

### 2.1. Fruit Morphological Development

*H. macrophylla* fruit is a capsule, composed of an outer pericarp, flesh, inner pericarp, and seeds. After pollination, the fruit’s morphology and color change with the growth process (Figure 1a). The fruit’s skin color is bluish purple at 0–20 Dap, light green at 30–40 Dap, dark green at 50–70 Dap, yellow-green at 80–90 Dap, and yellowish brown and dry at 100 Dap. The color of the stigma gradually changes from bluish -purple to pink, light green, and dark green, until it becomes withered and yellow. The growth curves of the longitudinal and transverse diameters of the fruit were consistent, showing a single S shape (Figure 1b). Fruit growth was divided into three stages. In the 0–10 Dap stage, the break line slope was small, and the longitudinal and transverse diameter growths were slow. During the stage of 10–55 Dap, the longitudinal and transverse diameters increased rapidly, ovary enlargement was oblong, and the fruit shape index was 1.29. During the 55–100 Dap stage, the transverse and longitudinal diameters reached about 90% for the ripe fruit, while the fruit shape index remained in the range of 1.11–1.29, and the fruit’s shape was oblong. The fruit’s vertical and horizontal diameter growth showed a “slow–fast–slow” trend.

### 2.2. Embryo Development at Different Embryo Ages

In order to further explore the growth and development of hybrid embryos at different developmental stages, fruits at 10–90 Dap were selected for paraffin section observation. Embryos were not formed at 10–20 Dap (Figure 2a,b). At 30 Dap, the cotyledon and embryonal axis did not begin to differentiate, and the embryo body was spherical (Figure 2c); at 40 Dap, the cotyledon and embryonal axis began to differentiate, and the embryo body was bilateral symmetrical and heart-shaped (Figure 2d); at 50 Dap, the cotyledon and embryonal axis began to differentiate, and the embryo body was elongated and torpedo-shaped (Figure 2e–i). The 60 Dap–90 Dap embryo remains a torpedo shape.

### 2.3. Analysis of Sequencing Data

A total of nine fruit samples were selected for transcriptome sequencing at the three stages of hybrid embryo development: globular stage (S1), heart stage (S2), and torpedo stage (S3). A total of 57.08 Gb of clean data was obtained, and the clean datum for each sample was 5.77 Gb. The percentage of Q30 bases is 93.73% or above (Appendix A). For each sample, the clean reads and hydrangeas reference genome HMA_r1.2_1 database (https://plantgarden.jp/en/list/t23110, accessed on 18 August 2023) were used to perform a sequence comparison, comparing efficiency between 86.77% and 88.93% (Appendix A). Principal component analysis and correlation test were conducted on nine samples, and the results show that the three biological samples in the three periods have high repeatability and a strong correlation (Figure 3a). The distance between the samples in each group was very close, and the three groups of samples were separated from each other, indicating good repeatability within the sample group and high inter-group classification (Figure 3b). Therefore, the sequencing data were reliable and could be used for subsequent analysis.

#### 2.3.1. Differential Genes Expression Analysis

*p* ≤ 0.05 and |log2FC| ≥ 1 were used as the screening criteria for differential gene expression at the three developmental stages. In groups S1_vs_S2, S1_vs_S3, and S2_vs_S3, there are 3359, 3803, and 3106 DEGs, respectively (Figure 4a). A Venn map was drawn for the differential genes of the three comparison groups, among which 332 were common differential genes, and 957, 918, and 695 differential genes were specifically expressed in the S1_vs_S2, S1_vs_S3, and S2_vs_S3 groups, respectively (Figure 4b). Therefore, it is speculated that large numbers of gene transcriptions and translations are activated during the heart embryo stage of seed embryo development.

#### 2.3.2. Functional Analysis of Differential Genes with GO Enrichment Analysis

To further understand the functional categories of these DEGs, we conducted a GO enrichment analysis of the 6253 DEGs identified. We selected the top 20 functions with the most significant enrichment levels that were present in all pairwise comparison groups from each of the three categories: biological processes (BPs), cellular components (CCs), and molecular functions (MFs) (Figure 5). In biological processes, DEGs are enriched in cellular processes, metabolic processes, and biological regulation. In cellular components, DEGs are enriched in cells and intracellular components. In molecular functions, DEGs are enriched in catalytic activity, binding activity, and transcriptional regulation activity.

#### 2.3.3. Differentially Expressed Genes Participate in Pathway Analysis

To further analyze the role of differentially expressed genes in germ development, the top 20 KEGG pathways with the most significant differentially enriched genes were identified. These pathways were analyzed separately for each of the three comparison groups: S1_vs_S2, S2_vs_S3, and S1_vs_S3. Phenylpropanoid biosynthesis, plant hormone signal transduction, the MAPK signaling pathway – plant pathway, and starch and sucrose metabolism were significantly enriched in the three comparison groups (Figure 6, Appendix A). A total of 58, 60, and 55 differential genes was up-regulated in the plant hormone signal transduction pathway. There were 57, 59, and 43 differential genes up-regulated in the MAPK signaling pathway – plant pathway. Therefore, it can be inferred that a large number of genes related to cell proliferation and differentiation and hormone signal transduction are activated during the heart embryo stage during germ development.

### 2.4. DEG Analysis of Biosynthetic Pathways and Signal Transduction Pathways of Plant Hormones

#### 2.4.1. Auxin Synthesis Pathway and Signal Transduction Pathway

Plants synthesize tryptophan to auxin through four pathways: the indole acetamide pathway (IAM), the indole pyruvate pathway (IPA), the tryptamine pathway (TAM), and the indole acetaldoxime pathway (IAOx) [32]. There are three core components mediating auxin signaling, namely auxin receptors (TIR1/AFBs), auxin blocking proteins (AUX/IAAs), and growth response factors (ARFs) [33]. DEGs involved in IAA biosynthesis were screened from transcriptome data, belonging to *ALDH*, *TDC*, *UGT74B1*, *YUC*, and *DAO* gene families (Figure 7a, Appendix A). The expression levels of two *YUC* genes (Hma1.2p1_0271F.1_g109005.gene and Hma1.2p1_0271F.1_g109020.gene) increased with embryo development. The expression levels of two *ALDH* genes decreased gradually (Hma1.2p1_0802F.1_g220550.gene and Hma1.2p1_0967F.1_g242275.gene). Differential genes related to IAA signaling belong to the *AUXI*, *AUX/IAA*, *ARF*, *GH3*, and *SAUR* gene families (Figure 8a,b, Appendix A), where AUXI/IAA, GH3, and SAUR are early response factors. One *AUX/IAA* (Hma1.2p1_0760F.1_g214260.gene), two *GH3* (Hma1.2p1_0905F.1_g235370.gene and Hma1.2p1_1023F.1_g248970.gene), and one *SAUR* (Hma1.2p1_0032F.1_g019970.gene) decreased with embryo development. As a transcriptional activator, the auxin response factor (ARF) mainly regulates the expression of downstream genes, and the expression of the three *ARFs* is up-regulated during the embryonic development of the *H. macrophylla* hybrid. They are *ARFs* (NewGene_21460, NewGene_21461, and Hma1.2p1_0209F.1_g089090.gene.).

#### 2.4.2. Cytokinin Synthesis Pathway and Signal Transduction Pathway

Isopentenyltransferase (ITP) converts AMP, ADP, or ATP and dimethylallyldiphosphate (DMAPP) to IP-nucleotides, which is the main synthetic pathway for the generation of cytokinin in plants, mainly derived tZ and iP and their carbohydrate-bound cytokinins [34]. Its signal transduction is mediated by continuous phosphoric acid transfer. During seed development, low concentrations of CK promote the enlargement of embryonic or cotyledon cells [35]. During Hydrangea hybrid embryo development, *CKX* expression decreased first and then increased (Figure 7b, Appendix A). *CKX* is a key enzyme in the degradation branch of the cytokinin synthesis pathway and regulates endogenous cytokinin levels by degrading cytokinin [36]. Therefore, it is believed that cytokinin activity is strong in the heart embryo stage. Four *AHPs*, three *B-ARRs*, and one *A-ARR* participate in cytokinin signal transduction (Figure 8a,b, Appendix A), among which the expression of one *AHP* (Hma1.2p1_0012F.1_g007340.gene) decreases with embryo development, and *AHP* is a negative regulatory factor in the CK signal transduction pathway.

#### 2.4.3. Abscisic Acid Synthetic Pathway and Signal Transduction Pathway

Active abscisic acid is synthesized in plant cells mainly through the carotenoid pathway, and ZEP, NCED, ABA2, and AAO3 are the key catalytic enzymes in the successive steps of synthesis [37]. In ABA core signal transduction, PYR/PYL/RCAR proteins bind to ABA and PP2C proteins to inhibit the phosphatase activity of PP2C, thereby activating the function of SnRK2 [38]. SnRK2 regulates seed development and promotes the accumulation of stored products in ABA signal transduction [39]. During the embryo development of the *H. macrophylla* hybrid, the expression of three *AAOs* and one *CYP707As* is up-regulated (Figure 7c, Appendix A), which promotes ABA synthesis by increasing the catalytic activity, while the expression of ABA1/ZEP is decreased. One *PYR/PYL*, six *PP2C*, two *SnRK2*, and one *ABF* were involved in ABA signal transduction (Figure 8a,b, Appendix A). Among them, the expression of *PP2C* (Hma1.2p1_0571F.1_g180710.gene) decreased and that of *SnRK2* (Hma1.2p1_0873F.1_g231230.gene) increased.

#### 2.4.4. Gibberellin Synthesis Pathway and Signal Transduction Pathway

GA is a plant hormone of diterpenoids, which can promote cells division and elongation, fruit development, seed germination, and flowering [40]. Seven differential genes involved in GA biosynthesis were screened from the transcriptomic data, among which *CPS*, *KO*, and *KAO* were involved in GA synthesis, while *GA20ox* and *GA3ox* were involved in the conversion of GA, which were also two dioxygenases that catalyzed GA metabolism at a later stage (Figure 7d, Appendix A). The expression of *GA3ox* was inhibited by GA feedback, while the expression of *GA20ox*, which caused GA inactivation, was activated by GA feedback [41]. The expression of *GA3ox* (Hma1.2p1_0129F.1_g060100.gene), which was the highest in the heart embryo stage and the lowest in the torpedo embryo stage. *GA20ox* (Hma1.2p1_0020F.1_g013915.gene), showed the opposite expression characteristics. In the gibberellin signal transduction pathway, the *DELLA* protein acts as a negative regulator to inhibit plant growth and development [42]. After the binding of active GA and *GID1*, the protein complex with *DELLA* is formed, and then the repression of the *DELLA* protein is no longer occurs [43]. During *H. macrophylla* hybrid embryo development, the expression of *GID1* (Hma1.2p1_0209F.1_g089280.gene) decreased, while the expression of *DELLA* (Hma1.2p1_1054F.1_g252590.gene) increased (Figure 8a,b, Appendix A).

#### 2.4.5. Jasmonic Acid Synthetic Pathway and Signal Transduction Pathway

Jasmonic acid is a kind of lipid hormone in plants, which not only plays an important role in regulating plant growth and development, but also plays an important role in resisting stress [44,45]. JA can directly regulate gene expression, and can also be catalyzed to form MeJA and other metabolites [46]. During *H. macrophylla* hybrid embryo development, 11 genes in the JA synthesis pathway were differentially expressed, including one *PLA2*, one *DAD1*, three *LOXs*, three *AOCs*, one *OPR3*, one *OPCL1*, and one *JMT*. Among them, two *LOXs* (Hma1.2p1_0686F.1_g203361.gene, NewGene_8992) and two *AOCs* (Hma1.2p1_1162F.1_g262740.gene, NewGene_13810) showed an upward trend during germ development (Figure 7e, Appendix A). JAs are important signaling molecules within and between plant cells. Under normal growth conditions, *JAZ* proteins bind to the *MYC2* transcription factor, collaboratively suppressing the expression of early JA-responsive genes, thus controlling the level of JA expression within the body [47]. With the development of seed embryos, the expression of five *JAZ* and two *MYC2* components is up-regulated (Figure 8a,b, Appendix A).

#### 2.4.6. Ethylene Synthesis Pathway and Signal Transduction Pathway

Ethylene is a bioactive gaseous plant hormone that regulates many physiological processes, such as seed germination, flowering, organ aging, and fruit ripening [48,49,50]. In the biosynthesis of ethylene, ACC is the most critical intermediate product, which is directly oxidized to ethylene [51]. Two *SAM* and three *ACO* differential genes participated in the synthesis of ETH, and the expression of *SAM* (Hma1.2p1_0052F.1_g030460.gene) showed a downward trend, while that of *ACO* (Hma1.2p1_0445F.1_g154290.gene) increased first and then decreased. It was highly expressed in the heart embryo stage (Figure 7f, Appendix A). The ethylene signaling pathway begins with mutual recognition, binding, and interaction between receptor proteins. The receptor is a negative regulator of ethylene signaling. During *H. macrophylla* hybrid embryo development, *ETR* (Hma1.2p1_0494F.1_g165480.gene) increased first and then decreased, with the highest expression in the heart embryo stage (Figure 8a,b, Appendix A).

#### 2.4.7. Salicylic Acid Synthesis Pathway and Signal Transduction Pathway

Salicylic acid is a plant hormone of small molecular phenolic compounds, which can regulate seed germination, fruit setting, and fruit ripening [52]. The phenylalanine ammoniase (PAL) pathway is the main pathway of SA biosynthesis [53]. Eight DEGs involved in the SA biosynthesis pathway were screened from the transcriptome data, and all of the eight *PAL* genes showed an upward trend during the development of the Hydrangea hybrid embryo (Figure 7g, Appendix A). The binding of class II TGA to the co-activator complex activates downstream gene transcription. During the development of the Hydrangea hybrid embryo, the expression levels of four *TAG* increased (Figure 8a,b, Appendix A).

#### 2.4.8. Brassinolide Synthesis and Signal Transduction Pathway

BR is an important phytosterol hormone that regulates many physiological activities, such as seed germination, cell elongation, and flowering time. Brassterol is synthesized in eight steps during the synthesis of brassinolide [54]. BRI1 protein is an important part of the BR signal transduction pathway, and the binding of BAK1 and BRI1 proteins can ensure the effective reception of the BR signal [55]. Two differentially expressed genes of *BRI1* (Hma1.2p1_0767F.1_g215530.gene and Hma1.2p1_2431F.1_g322035.gene) and *BAK1* (Hma1.2p1_0653F.1_g196100.gene) were selected from the transcriptomic data to perform in BR synthesis. The embryonic development of the Hydrangea hybrid showed an upward trend (Figure 8a,b, Appendix A). The expression of *DWF4* (NewGene_5077) was the highest in the heart embryo stage and significantly decreased in torpedo embryo stage. The expression of *CYP92A6* (Hma1.2p1_0291F.1_g114980.gene) was the highest in the globular embryo stage and decreased in the heart embryo stage (Figure 7h, Appendix A).

### 2.5. TFs Involved in the Growth and Development of Hybrid Fruits

During the development of hybrid fruit, many differentially expressed transcription factors are involved. The top 10, in terms of number, include 108 MYB, 106 bHLH, 99 AP2/ERF-ERF, 96 C2H2, 91 NAC, 79 WRKY, 78 MYB-related, 77 bZIP, 61 C3H, and 54 FAR1 (Figure 9a, Appendix A). From a pairwise comparison of samples from three different periods, it was found that there were 236 TFs in the S1_vs_S2 comparison group (147 up-regulated and 89 down-regulated), 264 TFs in the S1_vs_S3 comparison group (191 up-regulated and 73 down-regulated), and 251 TFs in the S2_vs_S3 comparison group (154 up-regulated and 97 down-regulated) (Figure 9c, Appendix A). Through K-means cluster analysis of differentially expressed transcription factors, six different expression trends were found (Figure 9b, Appendix A), among which category 6 (344 TFs) showed an upward trend, which may play a role in promoting the growth and development of *Hydrangea* hybrid fruits. Class 2 (276 TFs) and class 5 (215 TFs) showed a downward trend, which may inhibit the growth and development of *H. macrophylla* hybrid fruits. The specific role of these TFs in different stages of *H. macrophylla* fruit development needs to be further explored.

### 2.6. RT-qPCR Verification

In order to verify the accuracy of the results of RNA-Seq, 12 DEGs related to plant hormone biosynthesis, signal transduction, and transcription factors were randomly selected for RT-qPCR analysis (Figure 10, Appendix A). The results show that the RT-qPCR results of these 12 DEGs and the abundance expression of RNA-Seq data are consistent, indicating the reliability and accuracy of the RNA-Seq data in this study.

## 3. Discussion

Seed formation mainly goes through two processes: embryo/endosperm development and seed maturation [56]. The development of a dicotyledonous embryo starts from the first division of the fertilized egg and goes through the pro-embryo stage, globular embryo stage, heart embryo stage, torpedo embryo stage, and mature embryo stage [57]. The embryo development of the *H. macrophylla* hybrid only experiences the pro-embryo, globular, heart, and torpedo stages. From 50 days after pollination until the fruit is yellow, the embryo body remains torpedo-shaped, which is inconsistent with the development process of other dicotyledonous plant embryos. This may also be one of the reasons for the low hybrid seed germination rate in Hydrangea species. The reason may be that endosperm cannot provide sufficient nutritional support for the embryo’s morphological transformation [11].

Seed formation is also regulated by a variety of hormones. Plant hormone signal transduction ensures normal seed development by regulating the synthesis and catabolism of a series of proteins during embryonic development [58]. From fertilization to embryonic morphogenesis, cytokinin plays a dominant role. During seed development, the decrease of cytokinin concentration and the decrease of signal transduction pathway activity promote the enlargement of embryonic or cotyledon cells [13]. *CKX* irreversibly lyses free CK, thereby reducing the level of active CK. In maize, overexpression of *CKX* has been shown to decrease endogenous cytokinin levels [59]. In this study, the expression of *CKX* (Hma1.2p1_0579F.1_g182670.gene, Hma1.2p1_1194F.1_g265700.gene, and NewGene_12164) genes decreased first and then increased. These results indicate that the rapid cell division of the hybrid hydrangea embryo is mainly in the heart embryo stage.

Auxin is a key component of seed development, and IAA plays an important regulatory role in early seed development and the endosperm [8]. Previous studies have shown that the protein encoded by the *YUC* gene family can catalyze the direct conversion of indole pyruvate into IAA [33]. Excessive expression of *YUC* can lead to the excessive accumulation of auxin in plants and lead to defects in plant embryo development [60]. In this study, the expression of *YUC* (Hma1.2p1_0271F.1_g109005.gene and Hma1.2p1_0271F.1_g109020.gene) genes was up-regulated gradually during the growth of the embryo, and the torpedo stage exhibited the highest level, which may be one of the reasons for the torpedo shape of the hybrid embryo. *AUX/IAA* is a negative regulator of early response in the IAA signaling pathway, which can negatively regulate the abundance of the auxin response factor (ARF), thereby mediating various plant development processes [61]. *ARF* affects the formation of embryonic morphology by regulating cortical cell initiation and asymmetric division [62]. *AUX/IAA* (Hma1.2p1_0760F.1_g214260.gene) gene expression was down-regulated during the embryo development of the Hydrangea hybrid. The up-regulated expression of *ARF* (NewGene_21460, NewGene_21461 and Hma1.2p1_0209F.1_g089090.gene) promoted the endosperm accumulation of auxin to some extent, which may affect the transition of the embryo to maturity.

GA can regulate the physiological state of and biochemical changes in seeds, and the seeds themselves have the ability to synthesize GA. The *DELLA* protein is a major negative regulator of the GA signaling pathway, which restricts the process of the cell division cycle through the activities of TCP14 and TCP15 and stops embryonic development [63]. The overexpression of *GID1* in Arabidopsis thaliana can relieve the inhibitory effect of *DELLA* protein on plant growth and promote seed development [64]. During the embryo development process of Hydrangea hybrids, the expression level of the *DELLA* gene (Hma1.2p1_1054F.1_g252590.gene) gradually increased, while the expression level of the *GID1* gene (Hma1.2p1_0209F.1_g089280.gene) decreased. This suggests that the hybrid embryo’s development potentially halted at the torpedo stage, causing it to retain the characteristic torpedo shape throughout fruit maturation. Consequently, when mature seeds are sown, they exhibit a low germination rate. GA2-oxidase (*GA2ox*) is a key enzyme in the later stages of gibberellin biosynthesis, functioning as a negative regulator. It can convert active GA1 and GA4 in plants into inactive GA8 and GA34; GA3-oxidase (*GA3ox*) is the key enzyme regulating the synthesis of active gibberellins in the final step of the gibberellin metabolic pathway, responsible for catalyzing the inactive GA9 and GA20 into biologically active GA1 and GA4 [65]. In this study, the expression trends of *GA2ox* (Hma1.2p1_0020F.1_g013915.gene) and *GA3ox* (Hma1.2p1_0129F.1_g060100.gene) were also completely opposite, indicating that, during the embryo development process of Hydrangea, these two gibberellin oxidase genes work in coordination to control the content of active gibberellins in the body. ABA plays an important role in regulating storage proteins in seeds [13]. In a previous study by [66], *AAOs* were identified as essential genes for ABA biosynthesis, and upregulated expression of *AAOs* was observed during the seed development process in buckwheat. In this study, the gradual increase in the expression of *AAOs* (Hma1.2p1_1413F.1_g281850.gene, Hma1.2p1_1413F.1_g281850.gene, and NewGene_17841) indicates the promotion of normal embryo development in Hydrangea. In conclusion, the different expression patterns of differential genes in plant hormone biosynthesis and signal transduction pathways, as well as the regulatory imbalance of various hormone signaling pathways, may be the reasons for the formation of globular embryo, heart embryo and torpedo embryo, and the development of the hybrid embryo in the torpedo stage.

In addition to structural genes, which play an important role in fruit and seed growth and development, transcription factors also play a significant role in these processes [67]. The results of this study show that the MYB and bHLH families had the largest number of transcription factors, followed by AP2/ERF-ERF, C2H2, NAC, WRKY, MYB-related, bZIP, C3H, and FAR1. MYB plays a key role in plant morphogenesis, growth and development, and the synthesis of primary and secondary metabolites. Xie et al. [68] found in their research on dragon fruit that R2R3-MYB plays a key regulatory role in the fruit ripening process, and the inhibitory factor Hu MYB1 is involved in regulating fruit ripening by inhibiting genes related to betaine biosynthesis. RGE1 is a member of the bHLH gene family, and a functionally disabled RGE1 mutant will lead to embryo growth retardation, eventually resulting in the formation of small and wizened seeds [69]. Therefore, transcription factors also play a key role in the development of hydrangea hybrid fruit. It is speculated that fruit growth and development affect the expression of transcription factors and thus regulate the morphological changes in embryos. The specific regulation needs to be explored.

## 4. Materials and Methods

### 4.1. Experimental Material

The materials used in the experiment were collected from the Hydrangea germplasm resource garden at Hunan Agricultural University, using *H. macrophylla* ‘Otaksa’, with a round flower type as the female parent, and *H. macrophylla* ‘Coerulea’, with a flat-top flower type as the male parent for intraspecific hybridization. Using conventional water and fertilizer cultivation and maintenance management, the greenhouse’s temperature was maintained at 15–35 °C, humidity was maintained at 50–80%, and sufficient light was guaranteed.

### 4.2. Growth Index of Hybrid Fruits

Fruit shape and color changes were observed from the pollination day to 100 d (100 Dap) after pollination. 15 hybrid fruits were randomly selected every 5 days to measure the transverse and longitudinal diameters with vernier calipers, and 21 groups could be divided according to the measurement time. The measurement was repeated three times for each group, and then the fruit shape index was calculated [70] and the fruit growth curve was drawn.

### 4.3. Hybrid Embryo Growth and Development Morphology

Every 10 d after pollination, 10 hybrid fruits were placed in FAA fixation solution for 24 h, and the fixed materials were placed in 75%, 85%, 90%, 95%, 100%, and 100% ethanol for 4 h, 2 h, 1.5 h, 1.5 h, 1 h, and 1 h, respectively, and then dehydrated step by step. Then, add an alcohol benzene mixture for 20 min, xylene for 20 min, xylene for 20 min, paraffin for 2 h, paraffin for 2 h, and paraffin for 2 h for step-by-step transparency, wax dipping, embedding, slicing, sticking, dewaxing, and rehydrating, an HE dye solution (Safranine O-Fast Green Stain Kit) for staining, and then seal the slice. The slice thickness was 4 mm. The slices were flattened in 40 °C warm water in a KD-P tissue spreading machine (Jinhua Kedi Equipment Co., Ltd., Jinhua, China) and baked in a 60 °C oven. Finally, the slices were placed under an Olympus BX43F optical microscope (Olympus Corporation, Shinjuku, Tokyo, Japan) to observe the seed embryo structure.

### 4.4. RNA Extraction, Library Construction, and RNA Sequencing

Total RNA was extracted with a FastPure^®^ Universal Plant Total RNA Isolation Kit, and the purity and concentration of RNA were detected with Nano-Drop 2000 (Thermo Fisher Scientific, Waltham, MA, USA). The integrity of RNA was evaluated by agarose-gel electrophoresis. The qualified samples were sent to Biomarker Technologies for library construction, quantified by a Qubit 3.0 fluorescence quantifier and Qsep400 high-throughput analysis system. After the library was qualified, RNA-Seq was sequenced by the Illumina NovaSeq6000 sequencing platform using the PE150 mode.

### 4.5. Transcriptome Analysis

Raw reads were obtained by removing connectors and low-quality sequences from the original sequence data to obtain high-quality clean reads. The resulting high-quality Clean Reads were compared to the hydrangea reference genome, The version information for reference genome Hydrangea_macrophylla. HMA_r1.2_1. Genome (https://plantgarden.jp/en/list/t23110, accessed on 18 August 2023). According to the quantitative results of FPKM, differential genes (DEGs) were analyzed by gene ontology (GO) function analysis and Kyoto Encyclopedia of Genes and Genomes (KEGG) pathway analysis. In this experiment, Fold Change ≥ 2 and FDR < 0.01 were taken as the significant conditions for gene expression difference, and at least two of the three repeats in one treatment could be used for expression analysis. With P < 0.05 as the screening condition, hypergeometric test was used to compare KEGG database to find the significant enrichment pathway of differentially expressed transcripts and identify the main signal transduction pathways involved in differentially expressed genes.Log-normalized transcriptome data (Log_2_FPKM) were utilized for principal component analysis (PCA), correlation analysis, and heat map generation. GraphPad Prism 8.0 was employed to create a differential gene histogram for comparison groups. The online tool jvenn was used to generate area-scale Venn diagrams. Log2FC values from nine groups were applied in TBtools for data mapping. All genes annotated in regulatory pathways were marked, and their relative expression heat maps primarily demonstrated the responses of specific metabolic pathways during fruit development. TBtools was also used for heat mapping of DEGs within each module. Adobe Illustrator (version 2021) was used for typesetting differential gene heat maps and creating metabolic flowcharts.

### 4.6. RT-qPCR Assay

RT-qPCR assay was performed using the Cham Q Vniversal qPCR Master Mix assay kit with three replicates per sample. *β*-TUB was used as the reference gene of differentially expressed genes (DEGs). We used (http://www.oligoarchitect.com/ShowToolServlet, accessed on 21 December 2023) the website to design the RT-qPCR primers. The relative expression data of DEGs were normalized by the 2^−ΔΔCT^ method.

## 5. Conclusions

The growth of fruit from the intraspecific hybridization of *Hydrangea macrophylla* ‘Otaksa’ × ‘Coerulea’ exhibits a single sigmoid growth curve. At 30 Dap, 40 Dap, and 50 Dap, globular embryos, heart embryos, and torpedo embryos were formed, respectively, and the torpedo embryo was maintained at 50–90 Dap. In this study, a total of 12,933 differential genes was quantified during globular, heart-shaped, and torpedo-shaped embryo stages. Four pathways were significantly enriched: Phenylpropanoid biosynthesis, plant hormone signal transduction, MAPK signaling pathway–plant pathway, and starch and sucrose metabolism. Genes related to germ development include: in the cytokinin pathway, *CKX*; in the auxin pathway, *YUC*, *AUX/IAA*, and *ARF*; in the gibberellin pathway, *DELLA*, *GID1*, *GA2ox*, and *GA3ox*; and in the ABA biosynthetic pathway, *AAOs*. The MYB transfactor family and bHLH transcription factor have the largest number. This study enriches the research of Hydrangea hybrid fruit and is conducive to further understanding the molecular mechanism of the development of *Hydrangea macrophylla* hybrid fruit, and provides a biological basis for improving the excellent breeding standards of *Hydrangea macrophylla*.

## Figures and Tables

**Figure 1 ijms-25-07812-f001:**
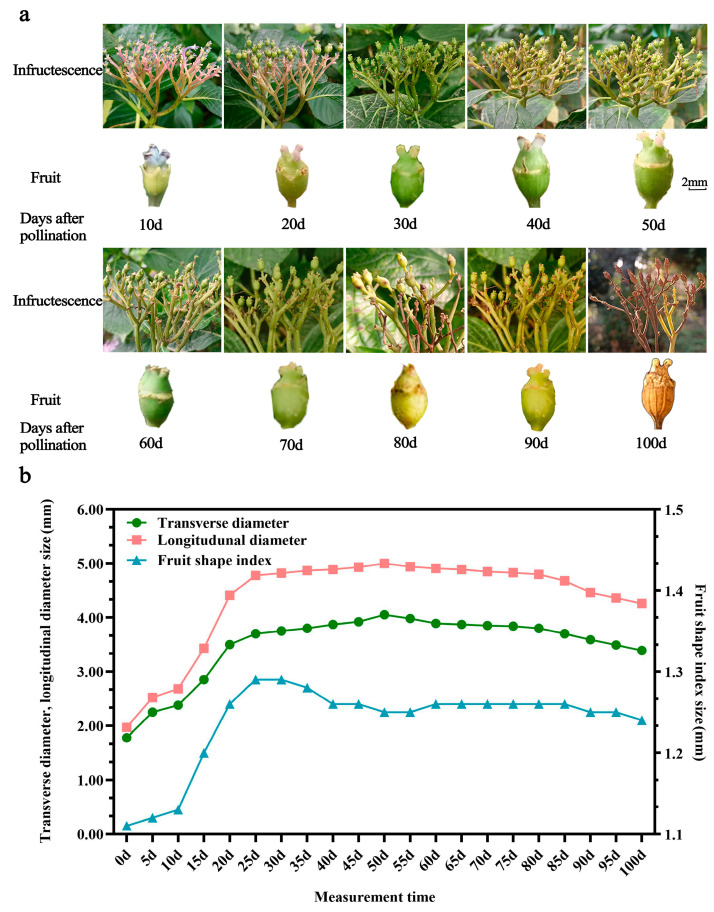
Morphological changes during the growth and development of *H. macrophylla* hybrid fruits. (**a**) Fruit shape changes; (**b**) fruit growth dynamic curve.

**Figure 2 ijms-25-07812-f002:**
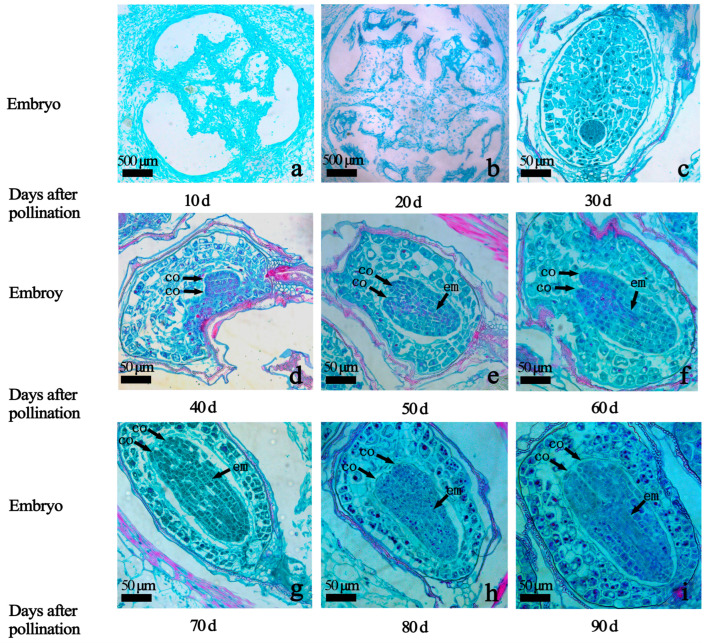
Morphological development of Hydrangea hybrid embryos: (**a**) 10 Dap, pro-embryo; (**b**) 20 Dap, pro-embryo; (**c**) 30 Dap, globular embryo; (**d**) 40 Dap, heart embryo; (**e**) 50 Dap, torpedo embryo; (**f**) 60 Dap, torpedo embryo; (**g**) 70 Dap, torpedo embryo; (**h**) 80 Dap, torpedo embryo; (**i**) 90 Dap, torpedo embryo. (Note: co: cotyledon; em: embryonal axis).

**Figure 3 ijms-25-07812-f003:**
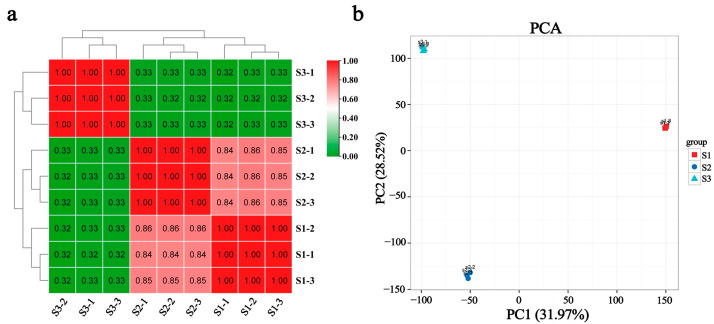
Biological repeated parallelism test. (**a**) Sample correlation analysis diagram; (**b**) PCA principal component analysis diagram.

**Figure 4 ijms-25-07812-f004:**
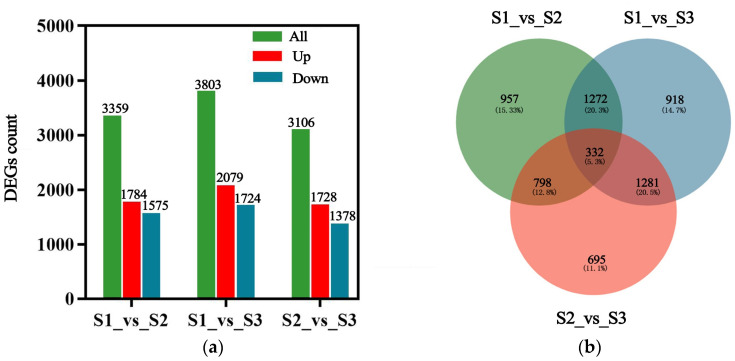
Differentially expressed genes at different stages of embryo development. (**a**) Differential gene expression statistics; (**b**) Venn diagram.

**Figure 5 ijms-25-07812-f005:**
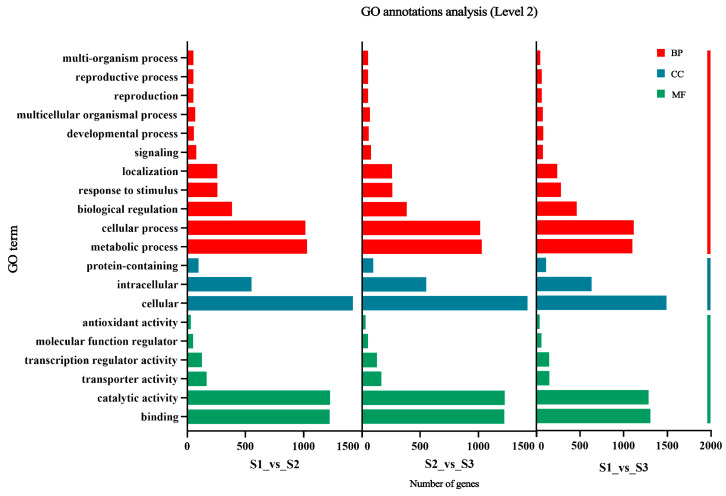
GO enrichment analysis of different genes in different developmental stages of hybrid fruits.

**Figure 6 ijms-25-07812-f006:**
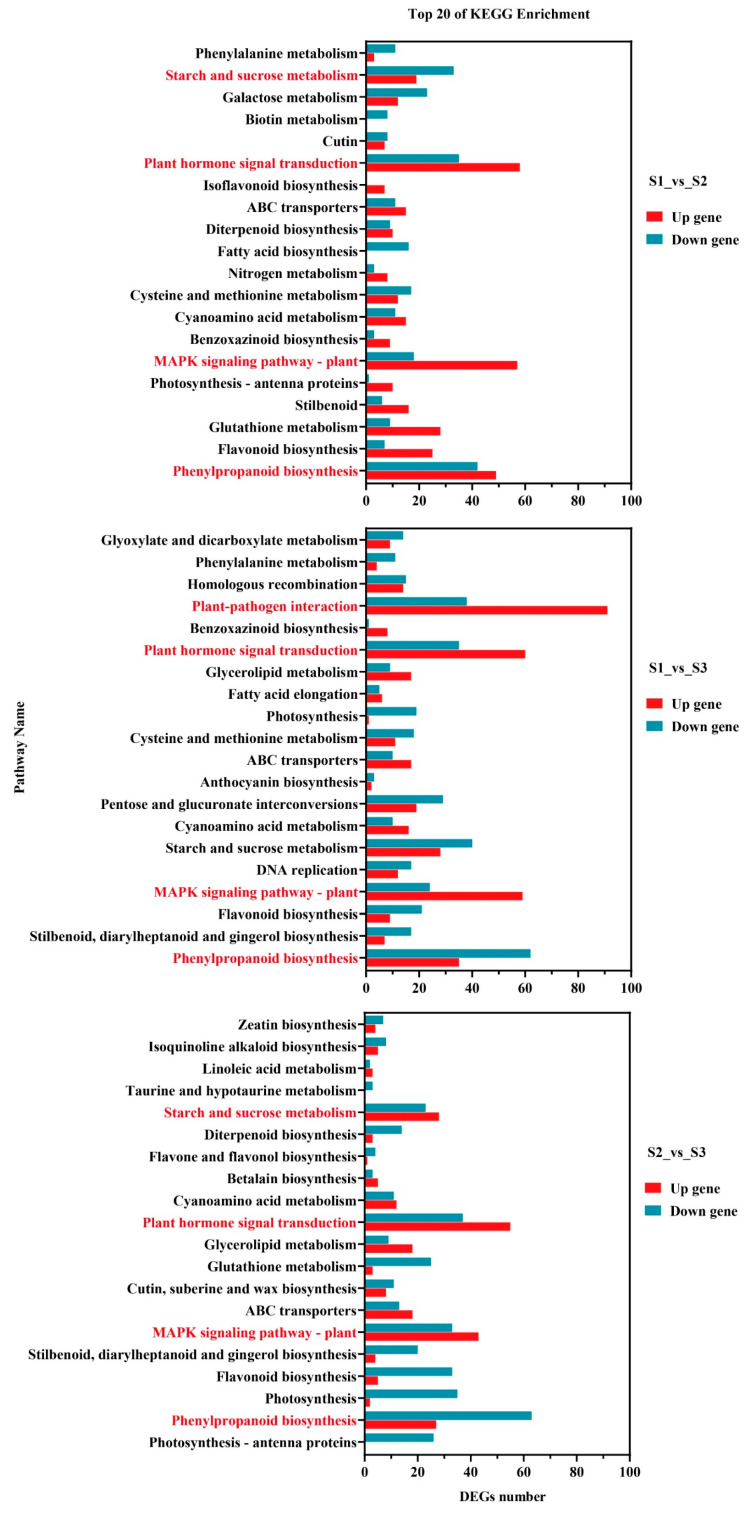
KEGG for full analysis of differential genes at different stages of embryo development.

**Figure 7 ijms-25-07812-f007:**
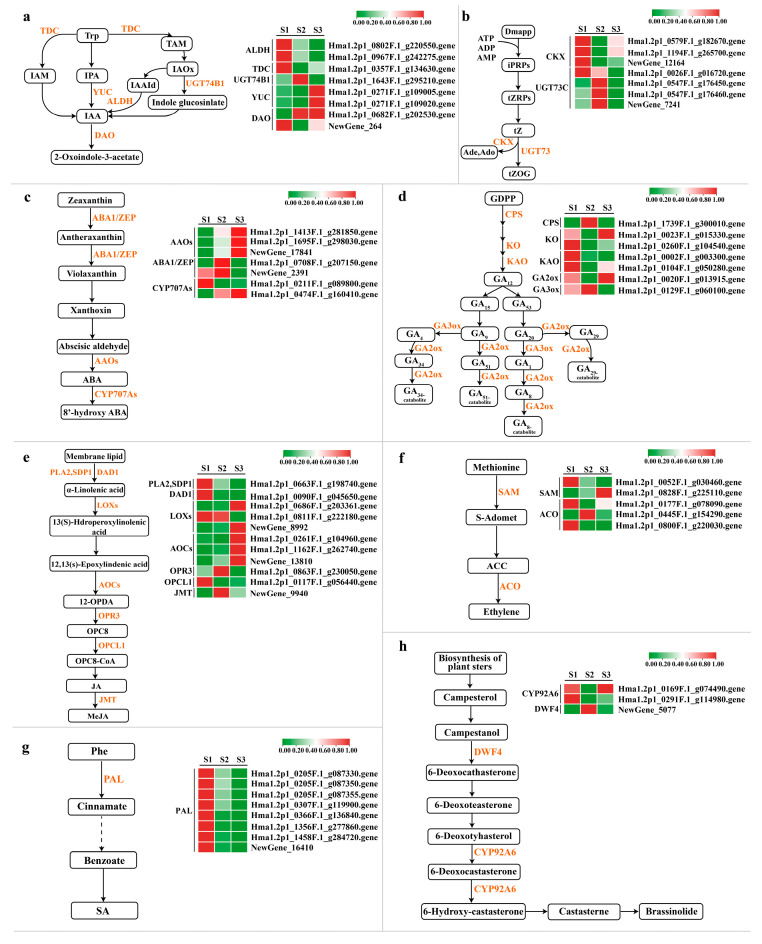
Gene expression of plant hormone biosynthesis pathway in the intraspecific hybrid fruit of *H. macrophylla*. (**a**) IAA biosynthesis pathway; (**b**) CK biosynthesis pathway; (**c**) ABA biosynthetic pathway; (**d**) GA biosynthetic pathway; (**e**) JA biosynthetic pathway; (**f**) ethylene biosynthetic pathway; (**g**) SA biosynthetic pathway; (**h**) BR biosynthetic pathway.

**Figure 8 ijms-25-07812-f008:**
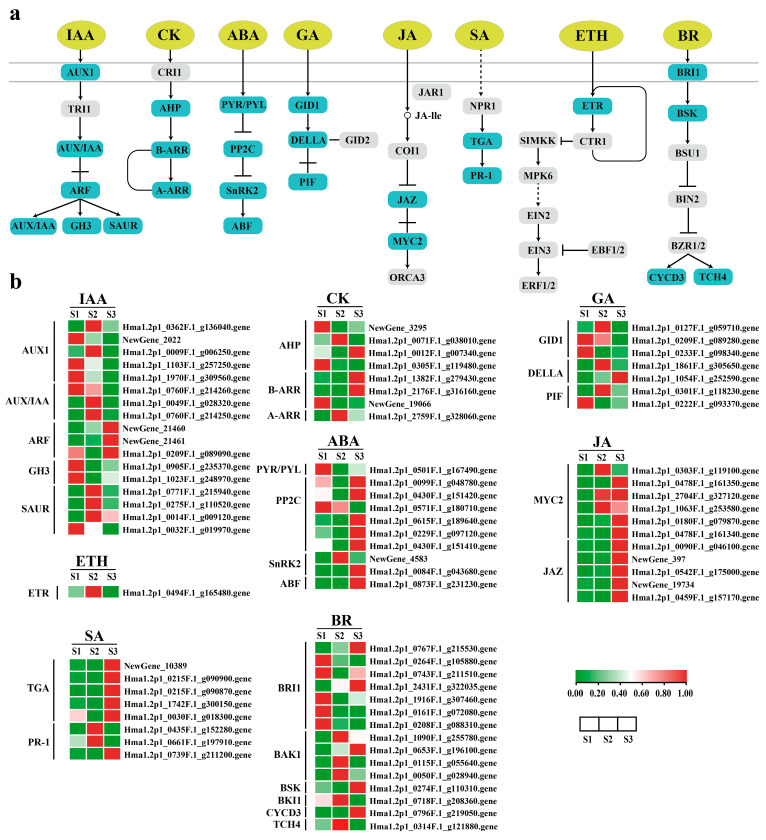
Gene expression of plant hormone signaling pathway in the seeds of intraspecific hybrids of *H. macrophylla*. (**a**) Schematic diagram of plant hormone signaling pathways; (**b**) heat map analysis of Z-score.

**Figure 9 ijms-25-07812-f009:**
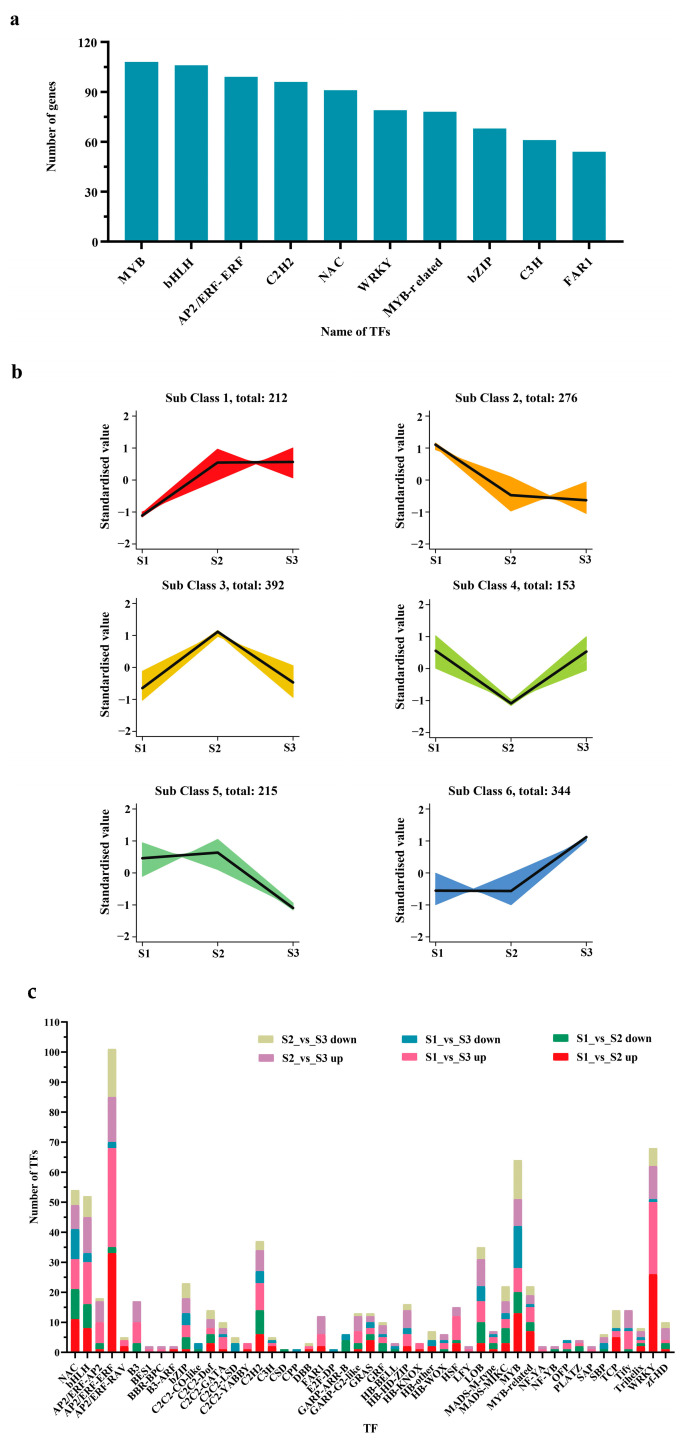
Characterization of transcription factors (TFs) expressed during the development of Hydrangea intraspecific hybrid fruits. (**a**) TF identification results at different developmental stages of Hydrangea intraspecific hybrid fruits; (**b**) K-means clustering analysis of 1592 TFs; (**c**) number of TFs differentially expressed at any two developmental stages of *H. macrophylla* hybrid fruits.

**Figure 10 ijms-25-07812-f010:**
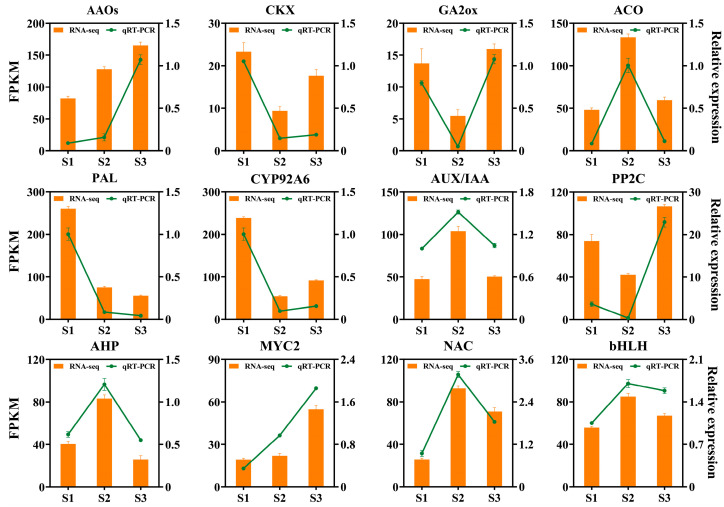
RT-qPCR was used to verify the selected genes in the RNA-Seq data. (Error lines identify three biological replicates).

## Data Availability

Transcriptome data are available at the National Center for Biotechnology Information (NCBI) database under the accession number PRJNA1085229 (https://www.ncbi.nlm.nih.gov/sra/PRJNA1085229, accessed on 7 June 2024).

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
