# Peer review of "Plant Hormone Pathway Is Involved in Regulating the Embryo Development Mechanism of the Hydrangea macrophylla Hybrid"

_ijms, 2024, doi:10.3390/ijms25147812_

Round 1
Reviewer 1 Report
Comments and Suggestions for Authors
This study showed that plant hormone pathway is involved in regulating embryo development mechanism of Hydrangea macrophylla hybrid.
The study M and M section is quite short but provides acceptable experimental designs. More details on the statistical analyses need to be included in the M and M section under Statistical analyses heading e.g. Correlation, PCA, gene expression statistics, Venn diagram, enrichment analyses, .
The results of this study contain some valuable elements. The study provides new results that several differential genes were quantified during globular, heart-shaped and torpedo-shaped embryos. Four pathways were enriched: Phenylpropanoid biosynthesis, Plant hormone signal transduction, MAPK signaling pathway-plant and Starch and sucrose metabolism. Several genes were shown related to e.g. germ development This study is help the further understanding the molecular mechanism of the development of Hydrangea macrophylla hybrid fruit.
Discussion provides limited comparisons of the results with previous studies. This needs to be improved a bit. For the conclusion, I suggest making bullet points to help the reader grasp the main points of this research. At the end of the conclusion, you should also address the limitations of your study.
References contains many inconsistencies e.g. mistakes in latin names, journal name abbreviation.
Overall, the study contains valuable results that can be considered for possible publication after suitable revisions.
Other suggestions:
- L9: This research is aimed to elucidate the role of plant hormones in …
- L9-37: Please reduce the brackets in the abstract
- L38: Give keywords that is not included in the title
- L47, L109, L117 and so on: H. macrophylla
- L208: Give KEGG in full.
- L551: Hydrangea serrata – and in italic
- L552: Hydrangea macrophylla – and in italic
- L570: Vitis vinifera – and in italic
- L573: Hordeum vulgare – and in italic
- L584: Triticum aestivum – and in italic
Reviewer 2 Report
Comments and Suggestions for Authors
Overall, the manuscript is very interesting, well-organized, and well-written. However, I believe it would be beneficial to improve the clarity and fluidity of the introduction. Moreover, it is necessary to enhance the Materials and Methods section, which lacks crucial information and clarity.
Please, check the entire text to ensure that the scientific names of species are written in italics.
Below are some specific notes.
Keywords: Replace the words already present in the title.
Introduction
Regarding the introduction, I suggest the authors revise the entire chapter, aiming to better integrate the literature discussion. As an example, I would rewrite the paragraph from line 57 to line 63 as follows, with each hormone discussed in a new line:
For instance, IAA produced by the central cell/endosperm of Arabidopsis mutants after fertilization promotes seed coat development [6]. Similarly, in maize, IAA levels were initially low during early endosperm development but increased gradually as the endosperm matured [7].
Literature results indicate that ABA has dual effects on seed embryo development. Specifically, increased ABA expression promotes embryo growth in early embryo development of Arabidopsis thaliana [8]. Conversely, decreased or insufficient ABA expression during tobacco seed development can lead to seed sterility and inhibit embryo growth [9].
L 115-119: This sentence doesn't make complete sense.
Results
Fig 1 a: Infructescence
Fig 1 b: insert space before (mm)
Fig. 2: Embryo. Moreover, check spaces in the caption
L 154: ‘A total of nine fruit samples were selected for transcriptome sequencing at the three stages of hybrid embryo development’ It should be highlighted and explained in the M&M, but it's not there.
L 160: ‘Principal component analysis and correlation test’ It should be highlighted and explained in the M&M, but it's not there.
Fig. 3: group S1, S2, S2?
L 173: Figure 4a
Fig 5: Number of genes
Discussion
L 380-386: Is this entire section referring to world literature? If so, additional bibliographical references are needed. Moreover, specific references to previous research, such as in this case on maize, should be better integrated into the discussion. For instance, in maize, overexpression of CKX has been shown to decrease endogenous cytokinin levels (46).
418: check grammar
422: GA34?
430: In a previous study by [54], AAOs were identified as essential genes for ABA biosynthesis, and upregulated expression of AAOs was observed during the seed development process in buckwheat.
439: check this sentence ‘In addition to structural genes play an important role in fruit and seed growth and development, transcription factors also play an important role in fruit and seed development.’
Materials & Methods
This section is the most deficient. In fact, there are several missing pieces of information. Therefore, the fruits used for sequencing analysis (9?) should be clearly specified, as well as the 3 groups analysed, since the entire study is primarily based on 3 different stages: S1, S2, and S3. Additionally, there are various details missing regarding data analysis. For instance, there is no mention of PCA, correlation, heat maps, etc.
L 461: was maintained
L 464. Color? There is nothing regarding the color determination.
L 467: group? Which?
L470-480: is not clear and must be rewritten. Moreover, Always use verbs in the past tense.
L 493: remove the full stop
L 494-499: seems like a repetition.
Line 503: replace with RT-qPCR assay
Comments on the Quality of English Language
English written form is good. I would improve the introduction.
Reviewer 3 Report
Comments and Suggestions for Authors
The paper presents a comparative transcriptomic analysis of hybrid fruits of Hydrangea macrophylla. Fruits containing embryos at the globular, heart, and torpedo stages were used. The author focused on the data analysis by describing the expression of genes that are related to the biosynthesis and signaling of plant hormones. Furthermore, the authors validate the expression of 12 genes related to this category. The paper is interesting and reasonably organized. However, I think it requires improvement in both the main text and supplementary information (see the comments below). I think the title should be reflect the findings reported. I feel that as it is now, it does not reflect the histology and transcriptomic data reported.
Comments:
1. Line 9-12. Double check the passage.
2. The main text needs to be free of typos. For example, there is a lack of space before the left square bracket in the references. Species names, for instance, line 61. Mutant names, for instance, line 81. Nouns, for instance line 229. Also line 244. The paper should be benefit from double check for typos along the text.
3. The introduction section should benefit from a brief description of the known data that explain the abortion of embryos in the species studied. Also, if there are previously reported or described Hydrangea macrophylla 'Otaksa' and Hydrangea macrophylla 'Coerulea', it should also be useful.
4. Section 2.1. Narrative for the use of hybrid fruits for the analys is required.
5. Line 154-160. This reference genome data is missing in the methods section. Specified accession sequenced (Otaske or Coerulea) or specified how close the sequenced specie is to the studied species in the text.
6. Line 160-162. Specified the statistical test used for correlation analysis.
7. Line 171-173. Double check the passage there is not c panel in the figure 4.
8. Figure 4. Properly labeled y-axis are required.
9. Figure 5. Software used for the GO analysis required. There is a typo in the label for the x-axis.
10. Line 199-202. I would suggest splitting the passage into two sentences to improve its readability.
11. Line 204-206. The link between cell proliferation and differentiation through GO analysis is unclear.
12. Section 2.4. I think the paper should be benefit from a premise in the main text that during the analysis presented in this section which threshold for DEG was used or if the analysis genes only required to be differentially expressed at least once a stage change over log2FC 1.
13. Line 224-226. It's not easy to follow the genes displayed as new genes in the supplementary table.
14. Lines 244-246, 281-283. I think the discussion section should host the speculative proposals.
15. Lines 256-258. Double-check the supplementary tables to harmonize the gene names with the gene names displayed in the figures.
16. Lines 300-302. Double check the number of MYC2 genes that have shown altered expression.
17. Lines 340-342. Double check the passage; does it refer to the top 10 largest TF families as DEGs? Also, double check the Table S2 (for instance, AP2/ERF-ERF number genes).
18. Lines 441-444. I think the conclusion should be close to the findings.
19. Lines 456-459. Does Otaksa lack pollen sources?
20. Lines 482-484. The definition of tissue collection for RNA-seq should be included in this section.
21. Section 4.5. Programs for processing raw data are required. Was the novo or mapped transcriptome approach utilized, as an example?
22. Lines 493-494. Double check the passage.
23. Lines 494-499. Repeated passage.
24 Section 4.5. The reference gene used is required. Does TUB mean tubulin reference gene? Primers need to provide in this section.
25 Lines 519-520. The passage needs to reflect the main findings.
26. Line 544. SUB14427261 is look that is not in the SRA.
All figures that display heat maps need to define the scale (Log2FC or Z-score).
References in lines 236-237, 248-252, 275-277, 291-292, 317-319, 326-330, 377-379, 383-385, 429-430, and 439-441 are required.
